# The Influence of Different Multipolar Mapping Catheter Types on Procedural Outcomes in Patients Undergoing Pulmonary Vein Isolation for Atrial Fibrillation

**DOI:** 10.3390/jcm13041029

**Published:** 2024-02-11

**Authors:** Kristof-Ferenc Janosi, Dorottya Debreceni, Botond Bocz, Dalma Torma, Mark Keseru, Tamas Simor, Peter Kupo

**Affiliations:** Heart Institute, Medical School, University of Pecs, 7624 Pecs, Hungaryy0v7wy@tr.pte.hu (M.K.);

**Keywords:** atrial fibrillation, pulmonary vein isolation, multipolar mapping catheter, catheterablation

## Abstract

(1) Background: During pulmonary vein isolation (PVI) for atrial fibrillation (AF), multipolar mapping catheters (MMC) are often used. We aimed to compare the procedural outcomes of two MMCs, specifically a circular-shaped and a five-spline-shaped MMC. (2) Methods: We enrolled 70 consecutive patients in our prospective, observational trial undergoing PVI procedures for paroxysmal AF. The initial 35 patients underwent PVI procedures with circular-shaped MMC guidance (Lasso Group), and the procedures for the latter 35 cases were performed using five-spline-shaped MMC (PentaRay Group). (3) Results: No significant differences were identified between the two groups in total procedure time (80.2 ± 17.7 min vs. 75.7 ± 14.8 min, *p* = 0.13), time from femoral vein puncture to the initiation of the mapping (31.2 ± 7 min vs. 28.9 ± 6.8, *p* = 0.80), mapping time (8 (6; 13) min vs. 9 (6.5; 10.5) min, *p* = 0.73), duration between the first and last ablation (32 (30; 36) min vs. 33 (26; 40) min, *p* = 0.52), validation time (3 (2; 4) min vs. 3 (1; 5) min, *p* = 0.46), first pass success rates (89% vs. 91%, *p* = 0.71), left atrial dwelling time (46 (37; 53) min vs. 45 (36.5; 53) min, *p* = 0.56), fluoroscopy data (time: 150 ± 71 s vs. 143 ± 56 s, *p* = 0.14; dose: 6.7 ± 4 mGy vs. 7.4 ± 4.4 mGy, *p* = 0.90), total ablation time (1187 (1063; 1534) s vs. 1150.5 (1053; 1393.5) s, *p* = 0.49), the number of ablations (78 (73; 93) vs. 83 (71.3; 92.8), *p* = 0.60), and total ablation energy (52,300 (47,265; 66,804) J vs. 49,666 (46,395; 56,502) J, *p* = 0.35). (4) Conclusions: This study finds comparable procedural outcomes bet-ween circular-shaped and five-spline-shaped MMCs for PVI in paroxysmal AF, supporting their interchangeability in clinical practice for anatomical mapping.

## 1. Introduction

Atrial fibrillation (AF) is known as the most common cardiac arrhythmia, affecting over 40 million people worldwide [1]. As per the recent guidelines from the European Society of Cardiology (ESC) on managing AF, the main purpose of opting for a rhythm control strategy is to ease AF-related symptoms and improve overall quality of life [1]. Catheter ablation for AF is deemed more effective than antiarrhythmic drugs (AAD) in sustaining sinus rhythm [2,3,4]. During catheter ablation treatment for AF, electrical isolation of the pulmonary veins (PV) is considered as the cornerstone of the procedure [1,5].

Multiple techniques exist for pulmonary vein isolation (PVI). The prevalence of AF ablations utilizing pulsed field ablation (PFA) techniques is on the rise [6,7], with available options including single-shot cryoenergy and radiofrequency (RF) devices [5,8]. Nevertheless, the globally predominant approach remains point-by-point radiofrequency (RF) ablation [8].

The guidance for point-by-point PVI procedures is facilitated by electroanatomical mapping systems (EAMS). After transseptal puncture, obtaining an electroanatomical map of the left atrium becomes pivotal for the ablation process.

MMCs are extensively employed in these procedures, offering supplementary insights into left atrium geometry creation, voltage mapping, complex fractionated atrial electrograms, validation of isolated pulmonary veins, and identification of reconnected or atrial fibrotic regions. Notably, they play a pivotal role in reducing both mapping and fluoroscopy time significantly [9,10,11,12]. Furthermore, their utility extends to facilitating the achievement of zero-fluoroscopy approach during PVI procedures [13].

Several multipolar mapping catheters are available in the clinical practice with different shapes, sizes, and electrode conformation, and most of them are widely used in cases of redo PVIs, left atrial focal tachycardias, and in macro-reentrant tachycardias; however, there is a lack of data referring the utilization of the PentaRay^TM^ NAV in de novo PVIs. In this prospective study, our objective was to assess and compare the procedural outcomes of two frequently employed mapping catheters integrated into the CARTO 3 EAMS (Biosense Webster Inc., Irvine, CA, USA) for PVI procedures. Specifically, we examined the PentaRay^TM^ NAV multielectrode catheter (Biosense Webster Inc., Irvine, CA, USA), characterized by five soft, radiating spines, and the circular-shaped LASSO^TM^ NAV catheter (Biosense Webster Inc., Irvine, CA, USA) which are equipped with 20 electrodes each (Figure 1).

## 2. Materials and Methods

### 2.1. Study Patients

In our prospective, observational trial, 70 consecutive patients undergoing PVI procedures for paroxysmal AF between November 2022 and July 2023 were enrolled. No sample size calculation was performed. Exclusion criteria encompassed (a) prior PVI procedures; (b) supplementary ablations extending beyond PVI, including both left and right atrial ablations; and (c) individuals below 18 years of age. We categorized the enrolled patients into two groups according to the type of MMC catheter employed during the ablation. The initial 35 patients, between November 2022 and March 2023, underwent PVI procedures with LASSO^TM^ NAV guidance (Lasso group). Subsequently, in cases 36–70, between April 2023 and July 2023, the PentaRay^TM^ NAV catheter was utilized for electroanatomical mapping due to the unavailability (i.e., backorder on the part of the manufacturer) of LASSO^TM^ NAV catheters (PentaRay group).

All procedures were conducted by the same expert electrophysiologist. The trial protocol adhered to the principles of the Declaration of Helsinki. The study protocol received approval from the Regional Ethics Committee (Approval No.: 9409/2022; Date: 18 November 2022). Written informed consent for participation was obtained from all patients.

### 2.2. Study Protocol

During the procedures, conscious sedation was induced using fentanyl ± midazolam after 12 h continuous fasting. Following local anesthesia, a decapolar steerable catheter (Dynamic Deca, Bard Electrophysiology, Lowell, MA, USA) was placed in the coronary sinus after vascular ultrasound-guided femoral venous puncture. Then, a single transseptal puncture guided by intracardiac echocardiography (ICE) was performed via SL0 (Abbott Laboratories, Chicago, IL, USA). From a distinct femoral venous puncture, the steerable 8.5-Fr-long sheath (VIZIGO, Biosense Webster Inc., Irvine, CA, USA) was directed to the superior vein cava, gently retracted, and secured against the intra-atrial septum. Subsequently, with the sheath’s guidewire penetrating the left atrium under fluoroscopic and/or ICE guidance, the VIZIGO was advanced over the initial transseptal puncture alongside the SL0 sheath. This sliding technique resulted in an SL0 and a VIZIGO sheath in the left atrium. Then, a MMC (either LASSO^TM^ NAV or PentaRay^TM^ NAV) was introduced into the left atrium via SL0. Additionally, a contact force (CF)-sensing radiofrequency (RF) ablation catheter (Navistar Thermocool SmartTouch ST NAV, Biosense Webster Inc., Diamond Bar, CA, USA) was positioned in the left atrium through a VIZIGO steerable sheath. A fast anatomical mapping of the left atrium was conducted with the MMC catheter, supported by the CARTO3 EAMS. No other mapping points were collected and analyzed other than an anatomical map. The ablation catheter operated in a power-controlled mode with a maximum power of 45 W for the anterior and 40 W for the posterior wall, employing a maximum temperature of 43 °C.

During RF ablations, the CARTO VISITAG™ Module was employed with a minimum stability time of 4 s and a maximum location stability range of 2.5 mm. The Visitag Surpoint (ablation index) was utilized with targets set at 350 for the posterior wall and 450 for the anterior wall. The target interlesion distance was maintained below 5 mm. The point-by-point ablation technique was applied, with real-time monitoring of CF and impedance. CF was maintained between 5 and 15 g during the ablation process.

To blind the operator from the presence or absence of first-pass isolation during ablations, the MMC catheter was positioned in the contralateral PVs. Intravenous unfractionated heparin was administered immediately after the femoral vein punctures, and an activated clotting time of >300 s was sustained throughout the entire procedure. The procedural endpoint of the ablation was considered achieved when all PVs were isolated.

### 2.3. Procedural Outcomes

The primary endpoint of this study was the procedure time, defined as the duration from the initial femoral vein puncture to the removal of the catheters. Additionally, various time intervals were compared, including the duration between femoral vein puncture and the beginning of mapping, mapping time, time between the first and last RF applications, validation time, and left atrial dwelling time. The first pass success rate, the number of RF applications, and the total RF time were also calculated. Mapping time was measured from the conclusion of the transseptal puncture until the initiation of the first RF ablation. Left atrial dwelling time was determined from the conclusion of the transseptal puncture until the withdrawal of sheaths from the left atrium.

Fluoroscopy time and radiation dose were automatically recorded by the fluoroscopy system. The RF generator (SMARTABLATE System, Biosense Webster Inc., Diamond Bar, CA, USA) documented the total number of RF applications and the overall ablation time.

The occurrence of major complications, such as vascular complications, pericardial effusion, cardiac tamponade, stroke, or atrio-esophageal fistula, was systematically assessed throughout the entire hospitalization and the periprocedural period.

### 2.4. Statistical Analysis

The data underwent analysis based on their conformity to normal distribution through the application of the Kolmogorov–Smirnov goodness-of-fit test. Continuous data were expressed using either the mean ± standard deviation (SD) or median (interquartile range, IQR), as deemed suitable. Categorical variables were represented by absolute numbers and corresponding percentages. Comparative assessments employed the chi-square test, *t*-test, and Mann–Whitney U test, as applicable. A significance threshold of *p* < 0.05 was employed for all statistical evaluations. The statistical analyses were executed using SPSS 28 software (SPSS, Inc., Chicago, IL, USA).

## 3. Results

Seventy patients were prospectively included. For the initial 35 patients, mapping and validation were conducted using a LASSO^TM^ NAV catheter (Group Lasso), whereas for patients 36–70, a PentaRay^TM^ NAV catheter was employed (Group PentaRay). No statistically significant differences were observed in the baseline characteristics of the study population between the two groups, including male sex distribution (Lasso: 80% vs. PentaRay: 74%, *p* = 0.57) and age (68.6 (58.7; 71.5) vs. 66.5 (50.6; 73.5), *p* = 0.36), as detailed in Table 1.

No significant differences were identified between the two groups in various procedural time metrics. Specifically, there were no differences in total procedure time (Group Lasso: 80.2 ± 17.7 min vs. Group PentaRay: 75.7 ± 14.8 min, *p* = 0.13). In addition, the time from the femoral vein puncture to the initiation of the mapping (31.2 ± 7 min vs. 28.9 ± 6.8, *p* = 0.80) was similar between the groups. Likewise, comparable findings were observed for mapping time (8 (6; 13) min vs. 9 (6.5; 10.5) min, *p* = 0.73), the duration between the first and last ablation (32 (30; 36) min vs. 33 (26; 40) min, *p* = 0.52), and the time required for validation (3 (2; 4) min vs. 3 (1; 5) min, *p* = 0.46). First pass success rates were also equal regardless the type of MCC used (89% vs. 91%, *p* = 0.71).

Regarding the left atrial dwelling time (46 (37; 53) min vs. 45 (36.5; 53) min, *p* = 0.56) and fluoroscopy data (time: 150 ± 71 s vs. 143 ± 56 s, *p* = 0.14; dose: 6.7 ± 4 mGy vs. 7.4 ± 4.4 mGy, *p* = 0.90), no significant differences were identified between the two groups. Additionally, the total ablation time (1187 (1063; 1534) s vs. 1150.5 (1053; 1393.5) s, *p* = 0.49), the number of RF ablations (78 (73; 93) vs. 83 (71.3; 92.8), *p* = 0.60), and total ablation energy (52,300 (47,265; 66,804) J vs. 49,666 (46,395; 56,502) J, *p* = 0.35) did not reveal any significant differences between the two groups. Results are summarized in Table 2.

## 4. Discussion

In our prospective, single-centre, observational trial comparing LASSO^TM^ NAV and PentaRay^TM^ MMCs, we identified no discernible differences in mapping, ablation, or fluoroscopy data among patients undergoing PVI for paroxysmal AF.

PVI is considered the gold standard technique in AF catheter ablation. While various ablation techniques can potentially achieve electrical isolation of the PVs, point-by-point RF PVI remains the most frequently employed method [8].

These procedures are guided by an EAMS, providing insights into both the left atrial anatomy and the localization of RF lesions [14]. In the context of point-by-point PVI procedures, a pivotal aspect involves generating the anatomical map of the left atrium, facilitated by either MMCs or the ablation catheter.

The initial experiences utilizing an MMC in PVI guided by EAMS were published in 2008, employing the PentaRay^TM^ NAV catheter with the EnSite EAM system [15]. Subsequently, the use of MMCs demonstrated superiority in PVI procedures compared to point-by-point contact mapping with the ablation catheter alone [10,16,17]. These advantages arise from the influence of interelectrode spacing and electrode size on mapping time and resolution. Smaller electrodes with closer interelectrode spacing can enhance mapping resolution and expedite mapping time. Bun et al. observed a quicker mapping time and the acquisition of more mapping points in left atrial tachycardia ablation with the PentaRay^TM^ NAV catheter compared to the conventional approach using the ablation catheter alone [17]. Moreover, in a study involving 30 patients with scar-related atrial arrhythmias, mapping with the PentaRay^TM^ NAV MMC improved mapping resolution in the scarred area compared to 3.5 mm electrode-tip linear ablation catheters [10]. Correspondingly, using the LASSO^TM^ NAV MMC proved beneficial compared to point-by-point mapping in patients undergoing repeat AF ablation procedures, particularly in left atrial scar mapping [16].

The novel 48-electrode, eight-spline design Octaray ^TM^ MMC (Biosense Webster Inc., Diamond Bar, CA, USA) has been shown to have an increased mapping speed and number of electrograms acquired, and was more accurate in identifying intact ablation lines compared to the PentaRay^TM^ NAV catheter in animal models [18].

The role of MMCs in the identification of atrial scar remains crucial; however, results in targeting low-voltage areas for a substrate-based ablation strategy beyond PVI are controversial [19,20,21,22,23,24,25]. In an ERASE-AF multi-center randomized clinical trial, Huo et al. found that PVI along with substrate modification was superior to PVI only in arrhythmia recurrences for the treatment of patients with persistent AF [19]; however, in a CAPLA randomized clinical trial including 338 patients undergoing catheter ablation for persistent AF, additional ablations beyond PVI did not significantly improve freedom from AF at a 12-month follow-up [23]. Furthermore, a systematic review and meta-analysis by Jia et al. including fourteen studies showed that patients in whom scarred atrial tissue was targeted had an even higher AF recurrence rate and did not have improved outcomes compared to PVI only [25].

In contrast to the LASSO^TM^ NAV catheter, the five-spline design of the PentaRay^TM^ NAV can be helpful in acquiring geometry in case of smaller pulmonary veins, which may be difficult to enter with the circular design of the LASSO^TM^ NAV. Furthermore, the splines of the PentaRay^TM^ NAV can visualize when the catheter is pressed against the wall of the atrium, helping avoid overestimation of the anatomy and acquiring a more accurate anatomical mapping. However, despite these advantages for the PentaRay^TM^ NAV catheter, it did not influence the procedural outcomes of PVI in our study.

## 5. Study Limitations

There are several limitations that need to be acknowledged. The trial was conducted at a single center, potentially limiting the generalizability of the findings to broader patient populations with varying demographic and clinical characteristics. The sample size of 70 patients might limit the statistical power, especially for detecting subtle differences between the two groups. Larger sample sizes would enhance the reliability of the findings and enable a more precise assessment of the comparability between the two catheter types. All procedures were performed by a single expert electrophysiologist, introducing the possibility of operator-specific influences on the outcomes. The results may not be universally applicable, and variations in operator skill and experience could impact the reproducibility of the findings in different clinical settings. The second group of patients, in which the PentaRay^TM^ NAV catheter was employed, were treated later than the LASSO^TM^ NAV group; therefore, it is not possible to rule out the potential for bias in terms of the operator’s experience. Finally, this study is not randomized, which could carry the potential for selection bias and difficulties in controlling confounding variables that might influence the outcomes.

## 6. Conclusions

In this prospective observational trial comparing circular-shaped LASSO^TM^ NAV and five-spline-shaped PentaRay^TM^ NAV catheters for PVI in paroxysmal AF, no statistically significant differences were detected in procedural times, first-pass success rates, or safety outcomes. These findings indicate comparable efficacy and safety profiles of the two catheter types, supporting their interchangeability in clinical practice for anatomical mapping during PVI procedures.

## Figures and Tables

**Figure 1 jcm-13-01029-f001:**
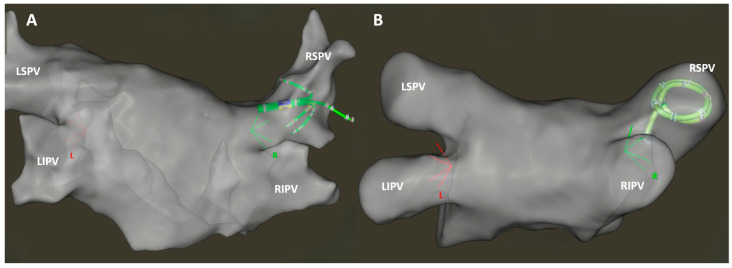
Anatomical map of the left atrium in a posteroanterior view, generated using the CARTO 3 electroanatomical mapping system. The map was created with the five-spline-shaped PentaRay^TM^ NAV catheter (**A**) and the circular-shaped LASSO^TM^ NAV multipolar mapping catheter (**B**). Both the LASSO and PentaRay catheters are positioned in the right superior pulmonary vein. Abbreviations: LIPV—left inferior pulmonary vein; LSPI—left superior pulmonary vein; RIPV—right inferior pulmonary vein; RSPV—right superior pulmonary vein.

**Table 1 jcm-13-01029-t001:** Baseline characteristics. Abbreviation: TIA—transient ischemic attack.

	Group Lasso (*n* = 35)	Group PentaRay (*n* = 35)	*p* Value
Male, *n* (%)	28 (80)	26 (74)	0.57
Age, y	68.6 (58.7; 71.5)	66.5 (50.6; 73.5)	0.88
Hypertension, *n* (%)	28 (80)	28 (80)	1.0
Heart failure, *n* (%)	5 (14.3)	6 (17.1)	0.74
Coronary artery disease, *n* (%)	5 (14.3)	8 (22.9)	0.36
Diabetes mellitus, *n* (%)	8 (22.9)	7 (20.0)	0.77
Chronic kidney disease, *n* (%)	6 (17.1)	7 (20.0)	0.76
Prior stroke/TIA, *n* (%)	1 (2.9)	5 (14.3)	0.09
Left atrial diameter, mm	54.5 ± 8.1	52.9 ± 7.8	0.18

**Table 2 jcm-13-01029-t002:** Procedural data and outcome. Abbreviation: NA—not applicable.

	Group Lasso (*n* = 35)	Group PentaRay (*n* = 35)	*p* Value
Procedure time, min	80.2 ± 17.7	75.7 ± 14.8	0.13
Time from access to start of mapping, min	31.2 ± 7.0	28.9 ± 6.8	0.80
Mapping time, min	8 (6; 13)	9 (6.5; 10.5)	0.73
Time between first and last ablation, min	32 (30; 36)	33 (26; 40)	0.52
Validation time, min	3 (2; 4)	3 (1; 5)	0.46
First pass rate, %	89%	91%	0.71
Left atrial dwelling time, min	46 (37; 53)	45 (36.5; 53)	0.56
Total ablation time, s	1187 (1063; 1534)	1150.5 (1053; 1393)	0.49
Number of ablations, *n*	78 (73; 93)	83 (71.3; 92.8)	0.60
Total ablation energy, J	52,300 (47,265; 66,804)	49,666 (46,395; 56,502)	0.35
Fluoroscopy time, s	150 ± 71	143 ± 56	0.14
Fluoroscopy dose, mGy	6.7 ± 4.0	7.4 ± 4.4	0.90
Complications, *n*	0	0	NA

## Data Availability

The data presented in this study are available on request from the corresponding author. The data are not publicly available due to Hungarian legal regulations.

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
