# Peer review of "The Influence of Different Multipolar Mapping Catheter Types on Procedural Outcomes in Patients Undergoing Pulmonary Vein Isolation for Atrial Fibrillation"

_jcm, 2024, doi:10.3390/jcm13041029_

Round 1

Reviewer 1 Report

Comments and Suggestions for Authors

The authors compared the circular-shaped LASSO catheter with the five-spline-shaped PentaRay catheter for establishing an electroanatomic map in 70 patients undergoing pulmonary vein ablation for paroxysmal atrial fibrillation. They found no differences in acute procedural outcome, such as procedure time. The authors conclude that both devices would be interchangeable.

While in general the manuscript is well written with adequate statistics, perfect English and adequate references, I question the design of the study. The authors write that it is a prospective study comparing both devices. However, the patients were not randomized. The first 35 patients were treated with the first device, while the latter was treated with the other. The authors did not report who selected the device in each individual patient. This means that we cannot rule out bias:

1.      In general, experience and technology get better with time. This means that the second device has a “headstart”: If the study would have shown a better outcome with the second device, this could also be attributed by time. Vice-versa, the current result does not rule out that the first device is better.

2.      The operators could have selected sicker patients for one device and healthier patients for the second device.

The authors should incorporate this major limitation into the Limitations section. Furthermore, they should report inclusion time (date of first vs. last inclusion).

The second problem with the study is that the author chose a “simple” procedure that does not require high-resolution electroanatomic maps, such as re-do procedures or ablation for atrial tachycardia. It is true that patients with PVI are most common and most comparable. Still, tiny differences between two mapping catheters may only be seen in more complicated procedures.

Lastly, it is unclear what the authors expected. The did not report a sample size calculation, which should be performed in a prospective study. If the authors used a convenience sample, the authors should report it.

Minor comment:

-        A part of the Introduction is written twice: lines 39-52 and 53-65.

-        Type of paper is missing (before title).

Author Response

The authors compared the circular-shaped LASSO catheter with the five-spline-shaped PentaRay catheter for establishing an electroanatomic map in 70 patients undergoing pulmonary vein ablation for paroxysmal atrial fibrillation. They found no differences in acute procedural outcome, such as procedure time. The authors conclude that both devices would be interchangeable.

While in general the manuscript is well written with adequate statistics, perfect English and adequate references, I question the design of the study. The authors write that it is a prospective study comparing both devices. However, the patients were not randomized. The first 35 patients were treated with the first device, while the latter was treated with the other. The authors did not report who selected the device in each individual patient. This means that we cannot rule out bias:

In general, experience and technology get better with time. This means that the second device has a “headstart”: If the study would have shown a better outcome with the second device, this could also be attributed by time. Vice-versa, the current result does not rule out that the first device is better.

We would like to thank for your valuable comment. All the procedures were performed by the same experienced operator, who has performed numerous PVI cases (>500 procedures) using both multipolar catheters before (Pentaray catheters for atypical flutter and redo PVI procedures) . However, in response to the Reviewer’s comment, we added the following to the Study limitations:

“The second group of patients, in which the PentaRay catheter was employed, were treated later than the LASSO group, therefore it is not possible to rule out the potential for bias in terms of the operator's experience.”

The operators could have selected sicker patients for one device and healthier patients for the second device.

Thank you for the comment. All the patients enrolled were cases for paroxysmal AF and based on the demographical data the two groups were homogenous as Table 1. shows, however we complemented the Study limitations with the following:

“Finally, this study is not randomized, which could carry the potential for selection bias and difficulties in controlling for confounding variables that might influence the out-comes.”

The authors should incorporate this major limitation into the Limitations section. Furthermore, they should report inclusion time (date of first vs. last inclusion).

Thank you for your comment. As requested by the Reviewer, we completed the Methods part with the inclusion dates.

The second problem with the study is that the author chose a “simple” procedure that does not require high-resolution electroanatomic maps, such as re-do procedures or ablation for atrial tachycardia. It is true that patients with PVI are most common and most comparable. Still, tiny differences between two mapping catheters may only be seen in more complicated procedures.

We would like to thank you for your valuable comment. PVI for paroxysmal AF is one of the most frequently performed catheter ablation. To date there is no scientific comparison between circular and five-spline shaped PentaRay catheters in terms of anatomical mapping only.  We agree with the Reviewer that some of the information provided by PentaRay catheters (e.g. high-density voltage maps) may not be essential for PVI only procedures, however the use of PentaRay multipolar catheters can still yield important data even in de novo PVI cases (e.g. amount of scar tissue can be prognostic for arrythmia recurrence) - DOI: 10.1016/j.jacc.2016.10.065

Lastly, it is unclear what the authors expected. The did not report a sample size calculation, which should be performed in a prospective study. If the authors used a convenience sample, the authors should report it.

Thank you for your comment. We did not use sample size calculation (i.e. convenience sample was used). We expanded the Methods accordingly.

Minor comment: 

 A part of the Introduction is written twice: lines 39-52 and 53-65.

Thank you for your observation. The duplicated part has been deleted and the mistakes listed has been corrected.

Type of paper is missing (before title).

Thank you for your observation. The missing title has been filled in according to the instructions of the journal.

Reviewer 2 Report

Comments and Suggestions for Authors

Janosi et al compared procedural outcomes during pulmonary vein isolation using EAM of two different types of mapping catheters in the Carto system. The authors reported no differences in procedural time, mapping time, left atrial dwelling time, and fluoroscopy time.

This is a potentially interesting article, but the presentation of the data and discussions need major revisions.

·       The introduction should better specify the usefulness of comparing these two mapping catheters.

·       Comparison of two mapping catheters would also find greater utility by reporting the number of points mapped between the two catheters (reporting catheter mapping density).

·       In the discussions, there is a large section on LVAs that is not strictly related to the article, whereas more emphasis should be placed on PVI only.

There is a section in the discussions where the effectiveness of mapping with the Orion catheter is reported, reporting its superiority compared to LASSO and PentaRay. I believe this is not a functional part of the topic of the manuscript and should be removed or edited

Author Response

Janosi et al compared procedural outcomes during pulmonary vein isolation using EAM of two different types of mapping catheters in the Carto system. The authors reported no differences in procedural time, mapping time, left atrial dwelling time, and fluoroscopy time. 

This is a potentially interesting article, but the presentation of the data and discussions need major revisions.

The introduction should better specify the usefulness of comparing these two mapping catheters.

We would like to thank you for the comments on how to improve our article. According to the Reviewer’s comment, to specify our purpose of the comparison, we added the following to the Introduction part:

Several multipolar mapping catheters are available in the clinical practice with different shapes, sizes, and electrode conformation, and most of them are widely used in cases of redo PVIs, left atrial focal tachycardias and in macro-reentrant tachycardias, however there is a lack of data referring the utilization of the PentaRay NAV in de novo PVIs.

 Comparison of two mapping catheters would also find greater utility by reporting the number of points mapped between the two catheters (reporting catheter mapping density).

Thank you for your comment. In our study, we focused solely on the acquisition of anatomical maps, without collecting any other mapping points or information related to voltage data or any other specific types of data. Confirming this, we have extended the Methods section

In the discussions, there is a large section on LVAs that is not strictly related to the article, whereas more emphasis should be placed on PVI only.

Thank you for the suggestion. According to the Reviewers suggestion, we rephrased parts of the Discussion to have more focus on the PVI itself and less emphasis on the LVAs.

There is a section in the discussions where the effectiveness of mapping with the Orion catheter is reported, reporting its superiority compared to LASSO and PentaRay. I believe this is not a functional part of the topic of the manuscript and should be removed or edited

Thank you for the comment. As requested, for the article to be more coherent, we removed the part regarding to the comparison of Orion catheter.

Reviewer 3 Report

Comments and Suggestions for Authors
Janosi et al. presents a paper "The influence of different multipolar mapping catheter types on procedural outcomes in patients undergoing pulmonary vein isolation for atrial fibrillation".
The paper is well written and the English form doesn't require major changes. However, what is missing, from my perspective, is the scientific resonance of this paper. It is extremely well-known that multipolar mapping catheters (high number of electrodes and reduced electrodes spacing) is of great importance in case of detailed substrate mapping (i.e., evaluation of scar) or in case of redo or in case of macroreentrant arrhythmia substrate-related. Having said that, in case of PVI only, the role of multi electrode high-density mapping is not extremely relevant and is also more expensive. For these reasons, I don't see what this paper can add to the current knowledges.

Author Response

The paper is well written and the English form doesn't require major changes. However, what is missing, from my perspective, is the scientific resonance of this paper. It is extremely well-known that multipolar mapping catheters (high number of electrodes and reduced electrodes spacing) is of great importance in case of detailed substrate mapping (i.e., evaluation of scar) or in case of redo or in case of macroreentrant arrhythmia substrate-related. Having said that, in case of PVI only, the role of multi electrode high-density mapping is not extremely relevant and is also more expensive. For these reasons, I don't see what this paper can add to the current knowledges.

We would like to express our appreciation for your comments and suggestions. PVI for paroxysmal AF is one of the most frequently performed catheter ablation. To date there is no scientific comparison between the circular and five-spline shaped PentaRay catheters in terms of anatomical mapping only. We agree with the Reviewer that some of the information provided by PentaRay catheters (e.g. high-density voltage maps) are not essential for PVI only procedures, however using Pentaray multipolar catheters can provide important data even in de novo PVI cases (e.g. amount of scar tissue can be prognostic for arrythmia recurrence). Regarding to the expenses, the price points of these catheters may vary from one center to another, to be more specific, in our institute there is no significant cost variance between the two catheters.

Round 2

Reviewer 1 Report

Comments and Suggestions for Authors

All comments have been addressed.

minor comment, please correct during final correction of the manuscript:

what does „other than“ mean at line 108 (red sentence)